# Research on Fault Detection by Flow Sequence for Industrial Internet of Things in Sewage Treatment Plant Case

**DOI:** 10.3390/s24072210

**Published:** 2024-03-29

**Authors:** Dongfeng Lei, Liang Zhao, Dengfeng Chen

**Affiliations:** College of Information and Control Engineering, Xi’an University of Architecture and Technology, Xi’an 710055, China; llldddfffldf@163.com (D.L.); zhaoliang@xauat.edu.cn (L.Z.)

**Keywords:** Industrial Internet of Things, deep learning, subsequence classification, fault detection

## Abstract

Classifying the flow subsequences of sensor networks is an effective way for fault detection in the Industrial Internet of Things (IIoT). Traditional fault detection algorithms identify exceptions by a single abnormal dataset and do not pay attention to the factors such as electromagnetic interference, network delay, sensor sample delay, and so on. This paper focuses on fault detection by continuous abnormal points. We proposed a fault detection algorithm within the module of sequence state generated by unsupervised learning (SSGBUL) and the module of integrated encoding sequence classification (IESC). Firstly, we built a network module based on unsupervised learning to encode the flow sequence of the different network cards in the IIoT gateway, and then combined the multiple code sequences into one integrated sequence. Next, we classified the integrated sequence by comparing the integrated sequence with the encoding fault type. The results obtained from the three IIoT datasets of a sewage treatment plant show that the accuracy of the SSGBUL–IESC algorithm exceeds 90% with subsequence length 10, which is significantly higher than the accuracies of the dynamic time warping (DTW) algorithm and the time series forest (TSF) algorithm. The proposed algorithm reaches the classification requirements for fault detection for the IIoT.

## 1. Introduction

In the Industry 4.0 era, the IIoT has become increasingly important to industrial production [1,2]. It is very useful to detect faults in a timely and accurate manner, which can help us more quickly identify the problems and take effective action. Various faults may have occurred during the operation of the IIoT [3], such as sensor disconnection, remote I/O offline, illegal system access [4], cyber-attacks, and so on.

Currently, there are several methods for fault diagnosis. Zhou, X. [5] realized the level-aware black-box adversarial attack strategy, targeting the graph neural network (GNN)-based intrusion detection in the IoT systems, with a limited budget. A. Pasyuk [6] provided an analysis and comparison of sequential feature selection methods for training machine learning models intended to classify network traffic flows. Yuri S [7] proposed a framework called Detection and Alert State for Industrial Internet of Things Faults (DASIF). Alberto G [8] proposed an approach to detect and classify faults that are typical in these devices, based on machine learning techniques that use energy, processing, and main application use as features. Jammalamadaka R K S [9] proposed an algorithm that uses deep learning techniques to forecast failures in smart home applications by analyzing each device’s log of events and calculating its failure rate per attempt. Qing Liu [10] proposed an innovative failure detection and diagnosis model for intelligent instruments in an IoT system using a Bayesian network, with a focus on handling uncertainties in expert knowledge and IoT monitoring information.

Anomalies in the IIoT are often caused by accidental factors, such as electromagnetic interference, network delay, system maintenance, sensor replacement, sensor sample delay, and so on. If each exception is to be handled, it will be time consuming and that reduces the capacity to handle the true exception. So, we planned to promote an algorithm to determine the fault by continuous abnormal points. Always, the IIoT gateway includes multiple network cards, such as the sensor data collection network card Ethernet 0 (Eth0), the system maintenance network card Ethernet 1 (Eth1), the point-to-point protocol card 0 (PPP0), and the virtual network card 0 (VPN0). We can acquire relevant information to evaluate the performance of the IIoT by analyzing the flow subsequences of the IIoT gateways.

At present, subsequence classification algorithms can be divided into four categories: (1) distance-based, (2) interval-based, (3) dictionary pattern-based, and (4) neural network-based subsequences. In terms of distance-based classification, Bagnall [11] proposed an algorithm called DTW [12] that adopts the KNN classifier for sequence classification. DTW requires defining a large number of subsequence models for pattern matching, which is always time consuming. In terms of interval-based classification, Deng H, Lines J, and Middlehurst M proposed the TSF, random interval spectrum ensemble [13], and typical interval forest [14] algorithms, which utilize statistical features such as the mean, variance, and slope of the subsequences for matching, and use random forest models for classification. Due to the large fluctuations in the numerical characteristics of the network flow subsequences, it is difficult to adapt the fixed patterns of statistical features to all types of subsequences. In terms of dictionary pattern-based classification, Lin J, Radford A, and He K, respectively, proposed a pattern packet algorithm [15], a symbol aggregation approximation algorithm [16], and a time series classification based on a word extraction algorithm [17]. These algorithms convert time series data into pattern packets and distinguish subsequence categories based on the relative frequency of a pattern packet’s appearance. Because the data span of the network flow is enormous, lots of patterns need to be defined. In terms of the neural network-based classification, Wang [18] and Fatwas [19] validated the performance of convolutional neural networks (CNNs) and residual neural networks in classification tasks [20].

For industrial IoT gateways, they always connect different amount sensors so that the flow features are variations that block the analysis of the flow sequence features. So, we encoded the flow sequence first, and then performed the fault diagnosis. Based on the above analysis, we proposed the IIoT fault detection algorithm SSGBUL–IESC. It consists of the SSGBUL module for sequence state generation and the IESC module for sequence classification. The main contributions of this work are as follows:We designed a code generator model, SSGBUL, to encode flow value and utilized the subsequence calibration function to reduce the prediction error during the encoding process.We identified the detail fault type by encoding the flow sequence. Firstly, we redefined the fault type tables by encoding sequences. And then, we converted the multi-dimensional flow sequences into one integrated code sequence. Finally, we identified the fault type using the integrated sequence and compared it with the encoding fault types.

## 2. SSGBUL–IESC Algorithm

The SSGBUL–IESC fault detection algorithm consists of the SSGBUL encoding module and the IESC classification module. There are three submodules within the SSGBUL encoding module: (1) the submodule of the network flow prediction based on unsupervised learning (NFPBUL); (2) the unified coding module (UCM) that is used to encode the network flow sequence; (3) the submodule of the calibrate the input subsequence (CIS). In the classification module, we combined the coding sequences of different network cards into one complete sequence. And then, we detected the fault type by comparing the complete coding sequence with the encoding fault type. Figure 1 shows the diagram of the overall network architecture.

### 2.1. SSGBUL Model

#### 2.1.1. NFPBUL Prediction Model

For a fixed network card of the IIoT gateway, the network flow is sampled within a specified time cycle according to Equation (1):(1)$$D=[d_{1},d_{2},\ldots ,d_{m}]$$
where *m* is the collection time and $d_{m}$ is the network flow amount. The network flow sequence *D* is divided into a collection $C_{d}$ with a sliding window length *l* according to Equation (2).
(2)$$C_{d}=\left\{\begin{array}{c} \left[d_{1}\;\;\;\;\;\;d_{2}\;\;\;\;\;\;\ldots \;\;\;\;\;d_{l}\;\;\right]\\ \left[d_{2}\;\;\;\;\;\;d_{3}\;\;\;\;\;\;\ldots \;\;\;\;d_{l+1}\right]\\ \;\;\;\;\;\;\;\;\;\;\;\;\ldots \\ {}[d_{m-l}\;\;d_{m-l+1}\;\;\ldots \;\;\;\;d_{m}] \end{array}\right\}$$

The label collection $C_{l}$ for the flow subsequence is constructed according to Equation (3).
(3)$$C_{l}={\left[d_{l+1},d_{l+2},\ldots ,d_{m}\right]}^{T}$$

The NFPBUL module consists of four parts: (1) the input layer, (2) the CNN layer, (3) the LSTM layer, and (4) the output layer. The input layer contains a set of network flow subsequences and a set of labels. The CNN layer has two parts: double one-dimensional convolutional components (1D-CNN) [21] and one max pooling component. The first convolutional component extracts the features from the input flow subsequence. The second convolutional component performs the extraction again to obtain an enlarged feature. The max pooling component simplifies the feature and uses it as an input in the decoder. These extracted features will be passed on to the LSTM layer to capture the long-term dependencies of the network flow. The LSTM layer consists of a single LSTM [22] component. The periodicity and regularity of the data are extracted through the LSTM [23] layer. The output layer contains two fully connected components. The first fully connected component is used to enhance the nonlinear ability of the LSTM model, and the second fully connected component is used to output the set of predicted values. Figure 2 shows the network structure of the NFPBUL.

We trained the NFPBUL model with the data from the subsequence collection $C_{d}$ and labelled the collection $C_{l}$ [24]. After the training was completed, the NFPBUL module was used for the prediction purpose. The calculation formula for network flow prediction is as follows:(4)$$C_{p}=N(C_{d},C_{l})$$
where the function N is the NFPBUL prediction model and $C_{p}$ is the prediction result set.

#### 2.1.2. Coding Model

According to Equation (2), the input sequence is constructed by using a sliding window on the test dataset, according to Equation (5).
(5)$$P_{t}=\{[d_{t-1-l},d_{t-l},\ldots ,d_{t-1}]\}$$
where $d_{t}$ is the network flow number. The future network flow can be predicted by the NFPBUL network, according to Equations (6) and (7).
(6)$$C_{p}=N(P_{t})$$
(7)$$p_{t}=C_{p}[0]$$

The UCM module encodes the network flow with the values of 1, 0, and −1 [25]. When the difference between the prediction value and the actual value is within the threshold ε, the value of the network flow is set to 0. When the difference exceeds the threshold ε, it means that several system faults have occurred, such as illegal access, cyber-attacks, etc., and the value of the network flow is set to 1. When the difference is below the threshold ε, several faults are predicted to occur, such as sensor disconnection, remote I/O module offline, etc., and the value of the network flow is set to −1. The calculation function of encoding is expressed in Equation (8):(8)$$F_{code}(d_{t},p_{t},\varepsilon )=\left\{ \begin{array}{l} 1,d_{t}\in [p_{t}+\varepsilon ,+\infty ]\\ 0,d_{t}\in [p_{t}-\varepsilon ,p_{t}+\varepsilon ]\\ -1,d_{t}\in [0,p_{t}-\varepsilon ] \end{array}\right.$$
where $d_{t}$ is the actual network flow at time t and $p_{t}$ is the prediction value at time *t*. The network flow coding sequence can be generated by multiple steps of predicting and encoding according to Equation (9).
(9)$$S_{t}=[s_{t},s_{t+1},\ldots ,s_{t+m}]$$

#### 2.1.3. Subsequence Calibration

At the NFPBUL model training stage, we always utilized the correct data. During the predicting stage, the NFPBUL model can predict the next data correctly for normal data. If one were to input abnormal data, the NFPBUL model will generate the future data according to the correct data trend. When the data returns to normal, the prediction value will still be generated by abnormal data, and that will lead to the system mis-encoding the normal data with abnormal data according to Equation (9). To resolve this issue, we proposed a subsequence calibration method to adjust anomalous data based on the difference between the actual value and the predicted value. The steps of this method are listed as follows:(1)Calculate the fixed position flow threshold value in a single data cycle on the training dataset according to Equation (10). Firstly, calculate the maximum network flow at each position. Then, subtract the average network flow to determine the error value $\varepsilon_{∆}$ according to Equation (10):(10)$$\varepsilon_{\Delta }=\max(d_{1*l},d_{2*l},\ldots ,d_{n*l})-\frac{\sum_{i=0}^{n}d_{i*l}}{n}$$
where $d_{i*l}$ is the flow value at a fixed position within the data circle.(2)Select the maximum threshold value as the whole sequence threshold according to Equation (11):
(11)$$\varepsilon =\max(\varepsilon_{\Delta }^{1},\varepsilon_{\Delta }^{2},\ldots ,\varepsilon_{\Delta }^{l})$$
where $\varepsilon_{∆}^{l}$ is the threshold at a fixed position within the data circle.(3)Calibrate the network flow. If the sequence is too regular, we can add a fixed value to ε prevent the model becoming too sensitive. Based on the difference between the actual value and predicted value, dynamically adjust the sequence item according to Equation (12). If the absolute difference value is greater than the threshold ε, it means that the actual value is abnormal, and construct the network flow subsequence with the predicted value. Otherwise, construct the network flow subsequence with the actual value according to Equation (12):
(12)$$a_{t}=\{\begin{array}{c} p_{t},|d_{t}-p_{t}|>\varepsilon \\ d_{t},|d_{t}-p_{t}|\leq \varepsilon \end{array}$$
where $a_{t}$ is the reconstructed network flow value at time *t*.(4)Obtain the prediction value using the NFPBTSN network model [26] through a newly constructed network flow subsequence [27] based on the calibration function according to Equation (13).
(13)$$p_{t+1}=C_{p}[0]=N(A_{t})=N([a_{t-l},a_{t-l+1},\ldots ,a_{t}])$$(5)Generate the network flow code sequence according to Equation (8).(6)Repeat steps 3 to 5 to generate the final code sequence after multiple rounds of prediction and encoding.

### 2.2. Classification Algorithm

#### 2.2.1. Integrated Module

We integrated multiple flow subsequences into a one-dimensional encoding sequence [28] that can display the whole IIoT gateway running information [29] according to Equations (14) and (15).
(14)$$S_{all}=\mathrm{Con}([S_{1},S_{2},S_{3},S_{4},S_{5},S_{6},S_{7},S_{8}])$$
(15)$$S_{i}=\mathrm{Con}([s_{1},s_{2},\ldots ,s_{l}])$$

In Equation (14), *S*_1_, *S*_2_, *S*_3_, *S*_4_, *S*_5_, *S*_6_, *S*_7_, and *S*_8_ represent the coding subsequence of the received or sent flow sequence. In Equation (15), $s_{l}$ is the encoded value of one point, *l* is the length of the sliding window, and Con is a connection function. Figure 3 shows the network flow sequences integrated diagram for Eth0, Eth1, PPP0, and VPN0 at the receiving and sending dimension.

#### 2.2.2. Encoding Fault Definition

Due to the varying quantity of sensors connected to the IIoT gateway, it is tedious to define the fault sequences for each gateway [30]. Thus, we proposed a new way to define the fault type by the encoding sequence [31] according to Equation (14) in Table 1. The encoding sequence includes eight subsequences, such as Eth0 receive subsequence, Eth0 send subsequence, Eth1 receive subsequence, Eth1 send subsequence, PPP0 receive subsequence, PPP0 send subsequence, VPN0 receive subsequence, and VPN0 send subsequence. The code of the lower position in the trend diagram in Table 1 is −1, which refers to the abnormal point where the flow value is lower than the normal value. The code of the higher position is 1, which refers to the abnormal point where the flow value is higher than the normal value. The code of the flat position is 0, which refers to the point where the flow value is normal. When the anomaly ends, the position of the point will return to a flat area from a lower or higher position. Currently, we can define 13 kinds of fault definitions, and can also append the new fault type.

According to the subsequence trend, the integrated code sequence is divided into linear subsequences and nonlinear subsequences. A linear subsequence means that the state values of each part remain relatively stable, such as the sensor disconnected exception, remote I/O fault, illegal system access exception, and cyber-attacks exception in Table 1. Nonlinear subsequences refer to sequences that undergo trend changes when one abnormal point begins or ends.

#### 2.2.3. IESC Classification Algorithm

We classified the fault type point by point [32]. There are two steps for the IESC classification algorithm. The first one is to obtain the integrated sequence for a point. And the second one is to classify the integrated sequence by comparing the integrated sequence with the fault type.

To obtain the integrated sequence, we chose the related encoding point according to the data point index. At the beginning, we obtained the different dimension encode values of the network cards by the data point index, such as Eth0 receive dimension, Eth0 send dimension, Eth1 receive dimension, Eth1 send dimension, PPP0 receive dimension, PPP0 send dimension, VPN0 receive dimension, and VPN0 send dimension. Next, we achieved the fixed length subsequences according to the sliding window by backtracking the dataset for each dimension. Finally, we combined different dimension encoding sequences into one integrated coding sequence that can display the IIoT gateway running information according to Figure 3.

For the classification stage, we compared the integrated encoding sequence with the fault definitions in Table 1 to obtain the exact anomaly category. At the beginning of the classification, we set the fault type −1 to mean fault unknown. During the comparing process, if one fault type had been matched, we then changed the fault type with the matched one. That can prevent the mis-adjustment problem when no fault type matched. The specific algorithm is shown in Algorithm 1.
**Algorithm 1. IESC Algorithm.**1: **Input**: integrated encoding sequence2: **Output**: fault type3: **Start**:4:5:   **function** compareSequence (sourceSequence, targetSequence)6:    flag ← 17:    **for** i = 1: sourceSequence.length **do**8:     **if** sourceSequence[i] != targetSequence[i] **then**9:      flag = 010:      **break**11:     **end if**12:     **end for**13:     **return** flag14:    **end function**15:16:   **function** IESC (inputEncodingSequence)17:     faultType ← −118:     **for** i = 1: faultList.size **do**19:     **if** compareSequence (inputEncodingSequence, faultList[i]) == 0 **then**20:      faultType = i21:      **break**22:     **end if**23:     **end for**24:    **end function**25: **End**

## 3. Data Acquisition

The datasets used in this article were sourced from a sewage treatment plant in 2023. The IIoT platform of the sewage treatment plant [33] consisted of four parts: sensors, remote I/O units [34], gateways, and a cloud server shown in Figure 4.

### 3.1. IIoT Architecture of Sewage Treatment Plant

As Figure 4 shows, sensors are used to sample the information of the devices and instruments such as thermometers, flow meters, water level meters, pH concentration meters, frequency converters, and so on. The remote I/O unit collects sensor data and provides the sensor data to gateways [35] based on the different industrial control protocols [36,37,38]. The IIoT gateway is responsible for sending sensor data to the cloud server via the MQTT format. The cloud server is used to analyze and display the sensor data.

### 3.2. Network Flow Collection Model

There are four different purposes of the network cards in the IIoT gateway: Eth0, Eth1, PPP0, and VPN0. Eth0 is used to collect sensor information from the remote I/O unit. Other industrial control protocols are converted to the Modbus TCP [39], providing a unified interface for data sampling. After data collection is completed, the sensor data will be transformed to the cloud server by the MQTT [40] protocol format through the PPP0 network card. System administrators can view system data through the VPN0 or Eth1 network cards. Figure 5 shows the network card functions.

### 3.3. Sensor Network

The first gateway was deployed at the sewage treatment workshop. The second one was deployed at the automatic dosing workshop. And the third one was deployed at the production workshop. The specific sensor connected information is shown in Table 2.

## 4. Experimental Results

We conducted four experiments based on three datasets from a sewage treatment plant: an ablation experiment, a linear fault detection experiment, a non-linear fault detection experiment, and an accuracy experiment with different lengths of sequence.

### 4.1. Dataset Introduction

Each dataset contains about 10,000 records. Figure 6 shows the flow sequences of several dimensions in Dataset1. The first half of each subgraph includes normal data that can be used to train the network model NFBUL. In the latter half of each subgraph, it includes the anomalies data that can be used for anomaly detection.

There are a different number of abnormal sequences in each dataset. Table 3 shows the abnormal sequence quantity in those datasets.

### 4.2. Typical Abnormal Sequence

Various types of failures may occur in the IIoT [41]. Several typical abnormal sequences [42] are listed below:Sensor disconnection. Sensor data are always sent to cloud servers in MQTT format. The content of MQTT includes data name and data value. Data value is obtained by converting different types of sensor values into character types, such as long, double, int, and so on. When this fault happens, the sensor data will become 0. So, the length of the converted MQTT transmission packet will be smaller than normal. And that will lead to the send flow amount of the network card PPP0 to decrease. Figure 7 shows the network flow diagram of sensor disconnection.Remote I/O offline. When this fault occurs, the IIoT gateway cannot collect sensor information connected to this remote I/O unit. So, the received network flow of the Eth0 will be decreased. Figure 8 shows the network flow diagram of the remote I/O offline fault.

Illegal access. When the system is being illegally accessed, the received and sent flow amount of Eth1 will increase a lot. Figure 9 shows the network flow diagram of illegal access. 

### 4.3. Experimental Metric

In this paper, we estimated the algorithm’s accuracy by comparing the original flow sequences of the sensor network and predicted flow sequences according to Equation (16). The annotate flow sequence is obtained by annotating the original subsequence of the sensor network, and the predicted flow sequence is obtained using the SSGBUL–IESC algorithm as follows:(16)$$P=\frac{1}{n}\sum_{k=0}^{n}Compare(\mathrm{Raw}(D_{t}),\mathrm{Detail}\_\mathrm{Classification}(D_{t}))$$
where $D_{t}$ is the original data subsequence, Raw is the function that obtains the annotation of the subsequence, and Detail_Classification is a function specific to the subsequence classification method.

### 4.4. Ablation Experiment

Figure 10a shows part of the Eth0 flow data in IIoT Dataset 1. During T1 and T2, one remote I/O unit disconnected, so the Eth0 receive flow decreased relatively. Figure 10(b.1) shows the coding result of the NFPBUL–UCM model. It can be observed that, during T1 and T2, the abnormal network flow is encoded as −1, 0, or 1. During T2 and T3, the normal network flow is encoded as 1 or 0. Figure 10(b.2) shows the encoding result of the SSGBUL model. We can see that all abnormal network flows during T1 and T2 are encoded as low threshold outliers −1, and the regular data are correctly encoded as 0.

To sum up, the NFPBUL–UCM encoding module has an incorrect coding problem. The SSGBUL encoding module generated the correct code within the subsequence calibration function when abnormalities occurred.

### 4.5. Compare Experimental Results

The length of the subsequence has a certain impact on fault diagnosis. In the experiment of linear subsequence classification and nonlinear subsequence classification, we set the subsequence length to 10.

#### 4.5.1. Linear Subsequences Classification

Figure 11 shows the fault detection accuracy for linear subsequences, such as sensor disconnection, remote I/O fault, illegal system access, and cyber-attacks.

From Figure 11, we can see that the linear fault detection accuracy of SSGBUL–IKNN is significantly higher than the accuracy of DTW and TSF.

Due to DTW’s inability to define enough subsequences for matching, the accuracy of identifying abnormal network flow subsequences decreases. Moreover, as the mathematical features of the network flow sequence cannot be efficiently extracted by TSF, the classification accuracy is reduced. Conversely, SSGBUL–IESC only needs to define the fault sequence based on the coding sequence of 1, 0, and −1, thus narrowing the scope of subsequence definition and improving the fault detection accuracy.

#### 4.5.2. Nonlinear Subsequences Classification

By identifying nonlinear subsequences, we can determine the start time or finish time of the anomalous. Figure 12 shows the accuracy of classifying nonlinear subsequences for the three algorithms.

As shown in Figure 12, the nonlinear subsequence classification accuracy for SSGBUL–IESC is significantly higher than DTW and TSF. When a cyber-attack or instance of illegal access occurs, the network flows of Eth1 and PPP0 change significantly, which is a big challenge for feature extraction. For DTW, the flow sequences are always outside of the matching subsequence models, which results in misclassifications. For TSF, the flow features are always beyond the boundaries of the original definition, which leads to decreases in accuracy. For SSGBUL–IKNN, the numerical sequences of network flow have been converted to code sequences, which can be compatible with various abnormal situations of data fluctuations.

#### 4.5.3. Different Subsequence Length Results

Table 4 shows the fault detection accuracies with different subsequence length, such as 5, 10, and 20.

We can find that with increasing the subsequence length, most of the accuracy in each dataset decreased a little, except the case for the subsequence length of 20 for Dataset 1, where due to some minor anomalies that existed in the dataset, our algorithm was not able to accurately detect. So, as the length of the subsequence increases, the accuracy of the algorithm’s detection decreases slightly. For the case of subsequence length of 20 for Dataset 1, it is because the total amount of sequence decreases when the length of sequence enlarges. So, we should prevent such long sequences in the application.

## 5. Discussion

Our initial focus was to study the continuous traffic characteristics of the IIoT for fault detection. At the beginning, we proposed the SSGBUL that is used to convert the flow data to code value. And then, we redesigned the classification module for flow sequence according to code fault definitions. Finally, we trained the SSGBUL model by the normal stage data in different datasets and verified the performance and compatibility for different IIoT gateways.

Our proposed method has certain advantages. Compared with the DTW algorithm, the code sequence only includes the values −1, 0, and −1, so the classification model becomes more efficient. Compared with the TSF algorithm, the statistical features of the code sequence are more apparent than the original network flow sequence. Therefore, the SSGBUL–IESC algorithm achieves the best fault detection results on the three IIoT datasets.

However, our proposed method does have several limitations. Firstly, this algorithm SSGBUL–IESC can only be used for specific flow datasets. These datasets only contain some features related to prediction, such as Modbus TCP, MQTT, and so on. In addition, the algorithm SSGBUL–IESC is sensitive to parameters, such as ε in the coding model, and the subsequence length in the IESC model. This will have a certain impact on the accuracy of the algorithm. Finally, the algorithm SSGBUL–IESC can detect limited quantity fault yet. Despite these limitations, all the results confirm that the SSGBUL–IESC algorithm can be successful for continuous abnormal sequence discovery for the IIoT.

## 6. Conclusions

Our research focused on fault detection by continuous abnormal sequence. We proposed a fault detection algorithm called SSGBUL–IESC based on unsupervised learning encoding. It effectively improves the accuracy and compatibility of fault detection in three IIoT datasets. The main results of the research in this paper are listed as follows:We designed a code generator model, SSGBUL, to translate the flow value to the unified code value and utilized the subsequence calibration function to reduce errors during the encoding process.We identified the detail fault type by encoding sequence type. Firstly, we redefined the fault type tables by encoding sequences. And then, we converted the multi-dimensional flow sequences into one integrated code sequence representing the operational status of the IIoT gateway. Finally, we identified the fault type by the integrated sequence by comparing it with the elements in the redefined fault type tables.

The experimental results show that the SSGBUL–IESC algorithm achieves an accuracy of over 90% with sequence length 10 on three IIoT datasets from a sewage treatment plant, thus meeting the requirements of IIoT applications.

In the future work, we will improve our algorithm to identify more types of faults.

In addition, we will further consider other factors of IIoT gateways for fault diagnosis, such as CPU usage, memory usage, system logs, and so on.

## Figures and Tables

**Figure 1 sensors-24-02210-f001:**
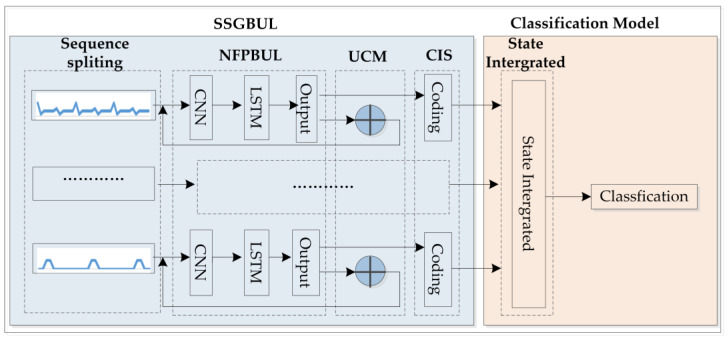
Overall network architecture of SSGBUL–IKNN.

**Figure 2 sensors-24-02210-f002:**
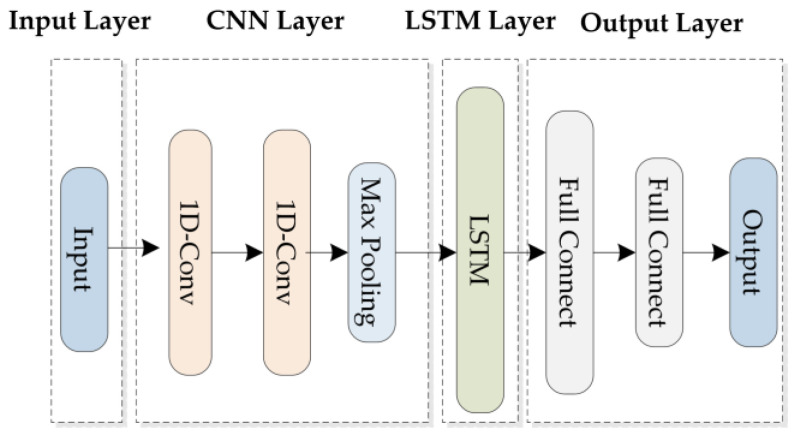
NFPBUL network structure diagram.

**Figure 3 sensors-24-02210-f003:**
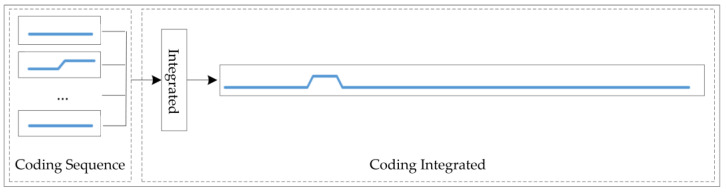
State of whole network card flow integration diagram.

**Figure 4 sensors-24-02210-f004:**
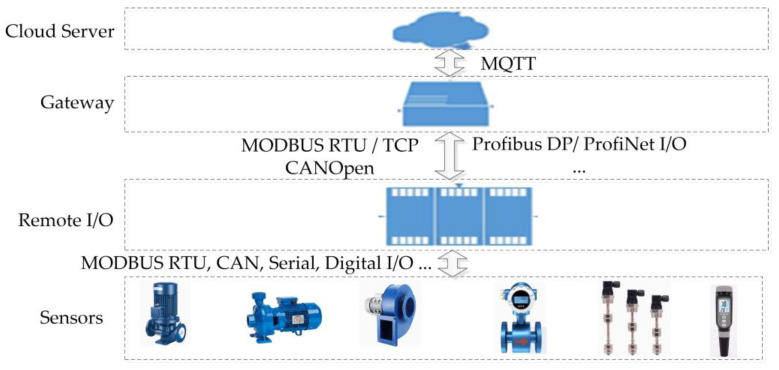
IIoT platform architecture diagram.

**Figure 5 sensors-24-02210-f005:**
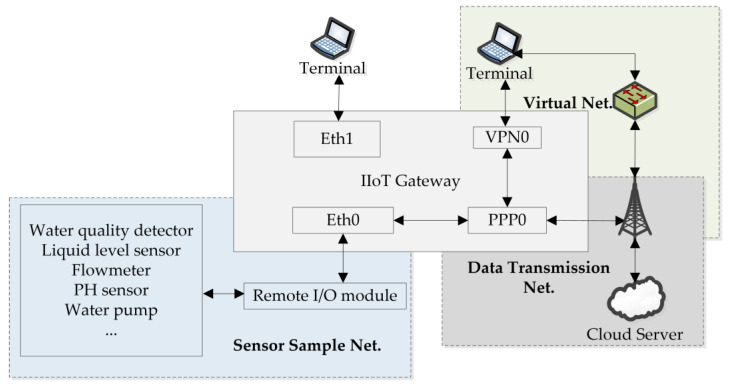
Function diagram for IIoT gateway network card.

**Figure 6 sensors-24-02210-f006:**
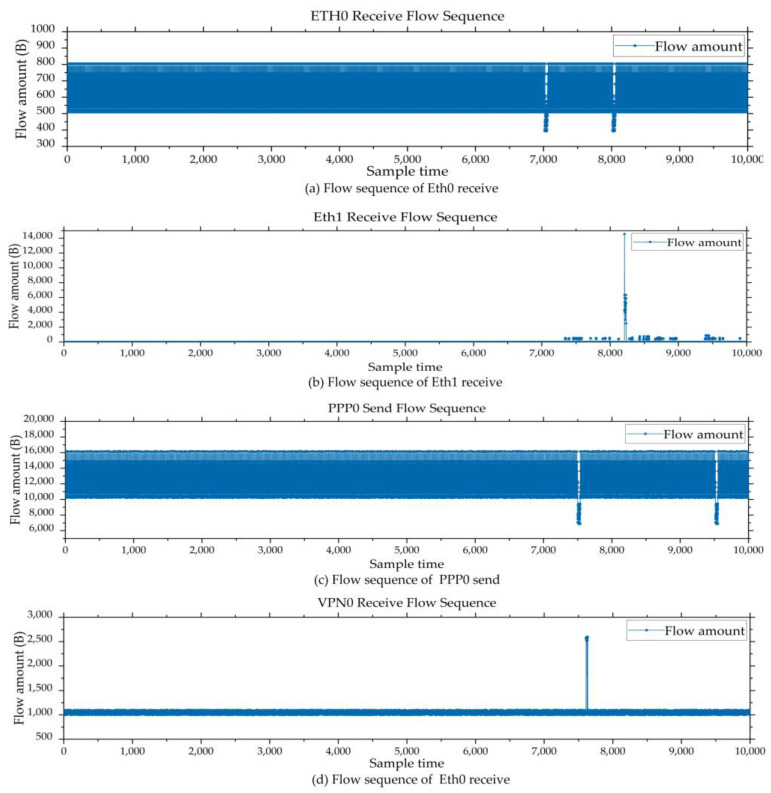
Dataset1 flow sequence diagram.

**Figure 7 sensors-24-02210-f007:**
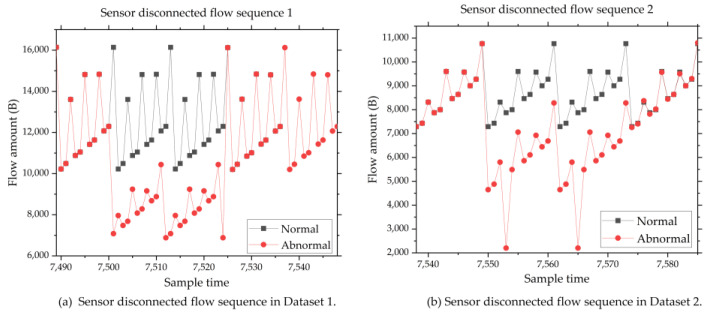
Network flow diagram for sensor disconnection.

**Figure 8 sensors-24-02210-f008:**
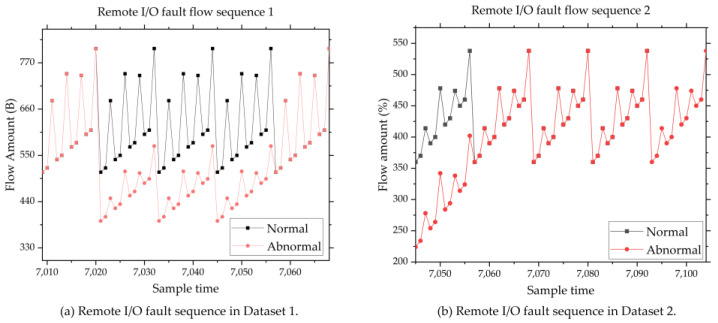
Network flow diagram for the remote I/O offline fault.

**Figure 9 sensors-24-02210-f009:**
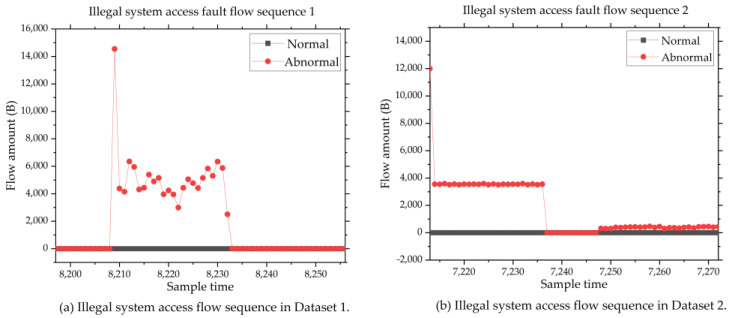
Network flow diagram of illegal access faults.

**Figure 10 sensors-24-02210-f010:**
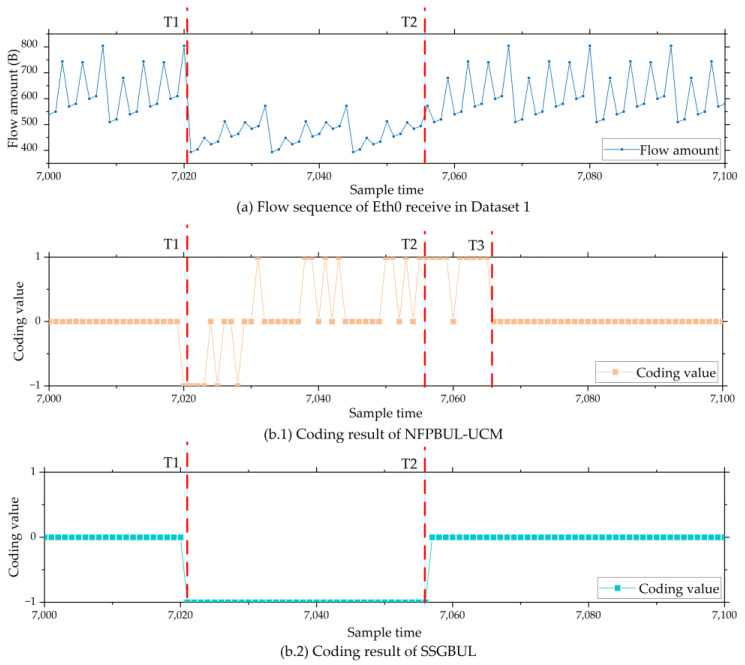
Encoding result diagram from NFPBUL–UCM and SSGBUL.

**Figure 11 sensors-24-02210-f011:**
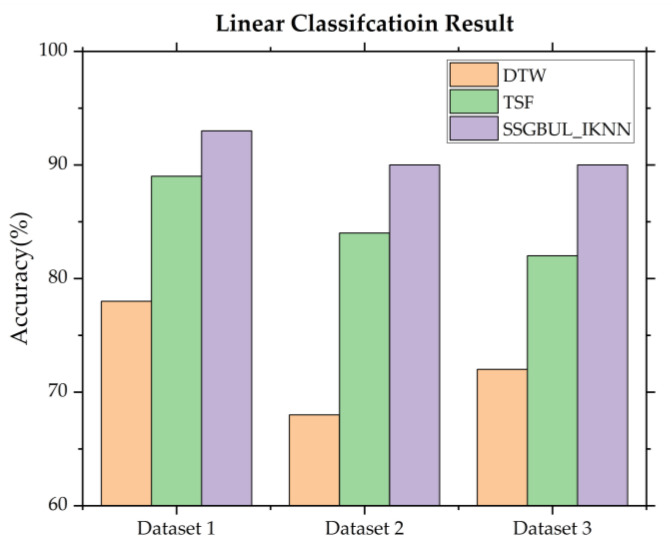
Linear fault detection classification accuracy.

**Figure 12 sensors-24-02210-f012:**
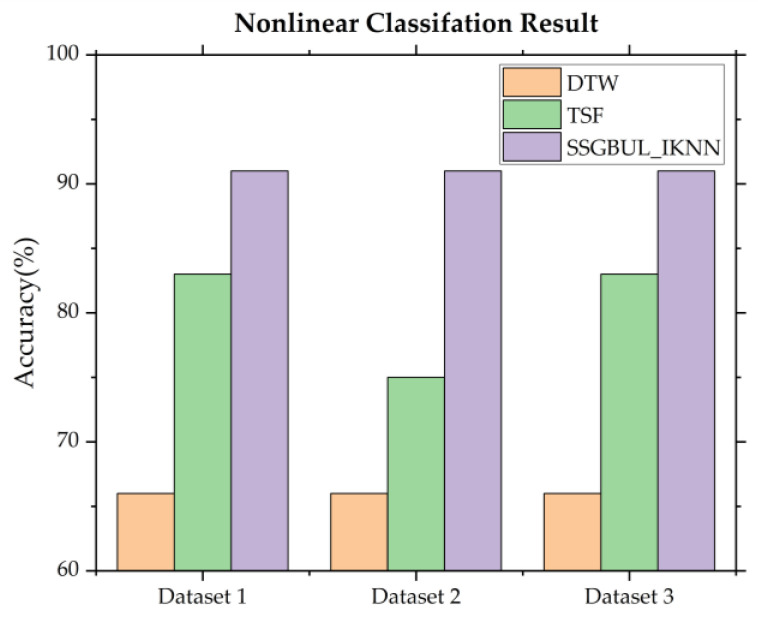
Nonlinear fault detection classification accuracy.

**Table 1 sensors-24-02210-t001:** Faults definition table based on coding sequence.

No.	Type	Trend Diagram
0	Normal	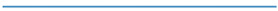
1	Sensor disconnected start	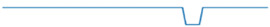
2	Sensor disconnected	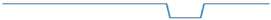
3	Sensor disconnected end	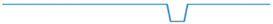
4	Remote I/O fault start	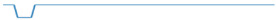
5	Remote I/O fault	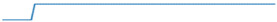
6	Remote I/O fault end	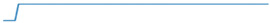
7	Illegal system access start	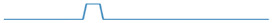
8	Illegal system access	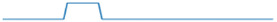
9	Illegal system access end	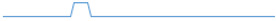
10	Cyber-attacks start	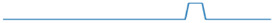
11	Cyber-attacks	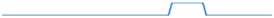
12	Cyber-attacks end	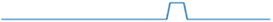

**Table 2 sensors-24-02210-t002:** Connected sensors amount for gateway table.

Gateway	Remote I/O Amount	Sensor Amount	Sensor Type
1	5	100	water level meter, frequency converter, water pump
2	3	55	pH concentration meter, flow meter, frequency converter, water pump
3	3	35	CO meter, CO_2_ meter, blower fan

**Table 3 sensors-24-02210-t003:** Abnormal sequences amount table.

Fault Type	Dataset1	Dataset2	Dataset3
Sensor disconnected	54	54	27
Remote I/O fault	54	66	81
Illegal system access	15	27	27
Cyber-attacks	15	27	15
Total	138	174	150

**Table 4 sensors-24-02210-t004:** Comparison table of accuracy with different subsequence length (%).

Subsequence Length	Dataset1	Dataset2	Dataset3
5	96.42	92.96	94.41
10	92.75	90.22	91.66
20	93.58	83.00	81.11

## Data Availability

The proposed dataset is private data and will be available upon request for research purposes.

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
