# Peer review of "Research on Fault Detection by Flow Sequence for Industrial Internet of Things in Sewage Treatment Plant Case"

_sensors, 2024, doi:10.3390/s24072210_

Round 1

Reviewer 1 Report

Comments and Suggestions for Authors

1. The title of the article is quite broad; however, the authors focused only on a few devices (from which they created their own research datasets) and did not utilize publicly available research datasets. Therefore, extrapolating their findings to the entire domain of IIoT seems unjustified. The title must thus reflect the authors' actual contribution to the addressed field.

2. The related work section describes various approaches to analyzing subsequences of time series, and the referenced scientific works have loose connections to the method presented in the article and the results obtained. However, numerous studies on anomaly detection in (Industrial) Internet of Things (I)IoT based on network traffic have already been conducted, numbering in the tens if not hundreds. Therefore, this section should reference more thematically related scientific articles.

3. The current form of the article impedes the verification and assessment of the authors' actual contribution to the field. In the Data Availability Statement, the authors wrote: "The data are sourced from a sewage treatment plant in 2023 and are contained within the article." However, this statement is not accurate. I found no source data or links to datasets within the article. While it is permissible for authors not to operate on publicly available open data, given the journal's thematic focus derived from the title "Applied Sciences," which allows for the publication of successful practical applications of known or novel methods, the failure to publish research data or scripts implementing the proposed method precludes any scientific discourse. Neither the reviewers currently evaluating this article nor future researchers reading it (if it is published) will be able to verify the obtained results or, equally importantly, refine the proposed method. In short, the current form of the article is inconsistent with the manner of presenting research results prevailing in scientific discourse, nor does it align with the spirit of research reproducibility.

4. Therefore, for a thorough assessment of this article, the authors should provide access to the source data (preferably by publishing them in one of the data repositories such as Zenodo or Figshare) and make available scripts implementing the proposed method (for instance, in a GitHub or GitLab repository), accompanied by a readme.txt file detailing step-by-step the various stages of the conducted research. There is no impediment to including these links in the published article for the benefit of future readers.

Comments on the Quality of English Language

Minor editing of English language required

Author Response

Dear expert:

Thank you very much for your valuable time! Please review again, thanks!

Reviewer 2 Report

Comments and Suggestions for Authors

Dear Authors,

I found your paper “Research on Fault Detection by Flow Sequence Algorithm with Deep Learning Model for Industrial Internet of Things” relevant to the fields of industrial Internet of Things and machine learning, presenting a fault detection algorithm called SSGBUL-IKNN, which is based on the flow sequence of the sensor network to achieve higher fault detection accuracy, exceeding 90%. The article is quite well written, presenting results that are of both theoretical and practical importance. However, the paper itself does not strictly follow the Sensors journal template (https://www.mdpi.com/files/word-templates/sensors-template.dot ) and, therefore, does not provide some key information that readers expect to find:

1. Please, combine the Introduction section with the Related Work section into a single Introduction section, in order to clearly describe the current state of the art and cite the key publications, as required by the journal template.

1.1. At this moment, the Introduction states, that “Traditional fault diagnosis methods often focus on the anomalies of individual sensors, which often leads to misjudgments caused by occasional sensor issues. <…> In additionally, for different IIoT applications, it is difficult to label data points due to the varying number of sensors and remote I/O.” Please provide citations to justify these statements. Are these statements based on the findings of other researchers? Or is this your own experience?

1.2. While incorporating related works into the Introduction section, please clearly state the problem which is solved by you and other researchers. What are the limitations of previous studies? Why do you need to do your own research? You must define the purpose of the work and its significance. Basically, your introduction should describe the problem that is being solved, then present how this problem is solved by other scientists and why results of related studies do not suit your case, and, finally, describe how your proposal differs from previous studies.

2. The article lacks the Discussion section, where “authors should discuss the results and how they can be interpreted from the perspective of previous studies and of the working hypotheses. The findings and their implications should be discussed in the broadest context possible. Future research directions may also be highlighted. “

2.1. Please add the Discussions section and describe how your solution helps to solve the problem and compare your findings with the results of previous studies.

3. Please provide a more detailed description of Figure 1 (in Section 3), which, as I understand, is the main figure of the article. The lower part of the figure is not mentioned at all in the text.

4. Please explain all notations in equations, e. g. what is the meaning of d in Equations 1 – 3?

5. Acronyms/Abbreviations/Initialisms should be defined the first time they appear in each of three sections: the abstract; the main text; the first figure or table. When defined for the first time, the acronym/abbreviation/initialism should be added in parentheses after the written-out form.

5.1. At this moment, the Abstract contains KNN, DTW, TSF abbreviations, which are not explained. The main text also contains unexplained abbreviations like PH (maybe pH?), Eth0, PPP0, VPN0, etc.

6. Figure and table captions should be more specific and reflect the content of the figure or table.

Comments on the Quality of English Language

The quality of the English language is quite good. I found only minor spelling errors and typos, which can be easily corrected with spellcheckers, but some expressions are difficult to understand, and therefore I would suggest reviewing the article by a native English speaker.

Author Response

(The authors gave the same response as above.)

Reviewer 3 Report

Comments and Suggestions for Authors

The article is skilfully written and addresses an intriguing topic related to identifying malfunctions or cyber-attacks in IIOT sensor systems.

Paragraph 3 provides a detailed explanation of the introduced algorithm. However, certain concerns arise within this section:

1: The referee recommends improvements to Figure 1, noting partial duplication in Figure 2. Figure 1 consists of an upper and a lower part (divided into three sections). While the upper part effectively illustrates the two modules, SSGBUL and IKNN, the lower part's sections on CNN layer and Output layer prefigure the contents of Figure 2 (albeit with slight variations). A decision must be made to either remove these two sections or eliminate Figure 2. Additionally, also in the lower part of Figure 1, the third section related to IKNN training appears incongruent with the context. Clarity could be improved by separating this third section to create a new figure (which should then be described in the body of the text).

2: The clarity of Table 1 is lacking. The Trend Diagrams do not provide sufficient explicitness regarding the type of event. The temporal distinction between the diagrams at the event's onset and when it is in progress (level -1 or +1) is unclear. Moreover, the reason for a return to normality (level 0) on the diagram before the abnormal event concludes remains unclear.

3: The presence of two Figure 4s impacts the numbering of subsequent figures and their references in the text.

4: Concerning the "ablation experiment," the presented diagrams and the reason why intervals T1 and T2 in section (a) and section (b2) are different in section (b.1) are not clearly elucidated.

The referee contends that the article's conclusions are understated and suggests that they should be expanded to better align with the overall content of the article.

Author Response

(The authors gave the same response as above.)

Round 2

Reviewer 1 Report

Comments and Suggestions for Authors

Thank you for the corrections you made in the article.

Reviewer 2 Report

Comments and Suggestions for Authors

Dear Authors,

Thank you for addressing the comments provided. The quality of the paper is improved, therefore, I believe that it can be published.

Good luck in your further research!

Comments on the Quality of English Language

Only minor editing is required.